# Effect of Vacuum Packaging on the Biochemical, Viscoelastic, and Sensory Properties of a Spanish Cheese during Chilled Storage

**DOI:** 10.3390/foods12071381

**Published:** 2023-03-24

**Authors:** Inmaculada Franco, Verónica Bargiela, Clara A. Tovar

**Affiliations:** 1Food Technology Area, Faculty of Sciences, Campus Universitario As Lagoas s/n, University of Vigo, 32004 Ourense, Spain; 2Department of Applied Physics, Faculty of Sciences, Campus Universitario As Lagoas s/n, University of Vigo, 32004 Ourense, Spain

**Keywords:** vacuum-packaging, San Simón da Costa cheese, chilled storage, ripening, sensory, viscoelasticity

## Abstract

The unique qualities of Spanish cheeses, such as the San Simón da Costa (SSC) cheese, are protected by the Protected Designation of Origin (PDO) status. The technological importance of chilled storage at 4 °C of vacuum-packaged (V) and natural (N) (unpackaged) cheeses was examined. For this purpose, the physico-chemical, biochemical, mechanical (puncture tests), viscoelastic (oscillatory and transient tests) and sensory properties of V and N cheeses were compared and analysed. During chilled storage, the caseins in V cheeses did not undergo proteolytic reactions. Low temperature maintained a low intensity of proteolytic phenomena for up to 6 months. Lipolysis was more intense in the N than in the V samples. The moisture content decreased in the N cheeses during chilled storage, and thus, the casein matrix concentration and ionic strength increased, resulting in an increase in the gel strength (S) parameter and complex modulus (G*), and the conformational stability−high stress amplitude (σ_max_). The low and similar values of the n’ and n’’ exponents (mechanical spectra) and the *n* parameter (transient tests) indicated the high degree of the temporal stability of the cheese network in both the N and V samples, irrespective of storage time. Likewise, the similar values of the phase angle (δ) for the N and V cheeses during storage indicate energy-stable bonds in the SSC cheese matrix. The attributes of the oral tactile phase (firmness, friability, gumminess, and microstructure perception), mechanical parameters and viscoelastic moduli enabled the discrimination of the packaged and unpackaged cheeses. Cheeses chilled and stored without packaging were awarded the highest scores for sensory attributes (preference) by trained panellists.

## 1. Introduction

The San Simón da Costa (SSC) cheese has enjoyed Protected Designation of Origin (PDO) status since 2005 [1,2]. It is a traditional smoked cheese with a minimum ripening period of 45 days. It has a distinctive shape, such as a large spinning top with a flat base and a pointed tip. The SSC cheese is manufactured in different parts of Galicia (Northwest Spain). It has a typical aroma and flavour, with a smoky note derived from the use of birch wood during the ripening process. The SSC cheese texture is fine, fatty, semi-hard, semi-elastic and dense; it has a creamy yellow colour and cuts smoothly. The outer layer of the cheese, 1 to 3 mm thick, is a yellow–ochre colour on the surface and somewhat oily. The edible part of the cheese is of fine texture, fatty, semi-hard, semi-elastic and dense, and between white and yellow.

SSC cheese predominantly undergoes enzymatic coagulation. The rennet-induced coagulation of milk is in fact a two-stage process: the primary phase involves the specific enzymatic modification of the casein micelles to produce paracasein micelles that aggregate in the presence of cations Ca^2+^ at temperatures above 20 °C. The aggregation of the rennet-altered micelles is referred to as the secondary phase of coagulation [3]. Specifically, para-casein submicelles are dissociated into their constituent proteins during processing and are subsequently incorporated into the water–fat interphase and the gel network [3,4].

The paracasein micellar matrix contains fat globules as a dispersed phase (filler). This is a common characteristic of composite gels [5], which can be considered as filled gels [6]. Thus, the fat filler contributes positively to cheese texture and rheology. Considering both physicochemical factors (pH and ionic strength) and biochemical reactions (proteolysis and lipolysis), different gel properties can be determined using two rheological methods based on different physical principles of measurement. The first analyses cheese as a viscoelastic gel within the linear viscoelastic range (LVER) and provides information related to the cheese microstructure [7]. The second is a mechanical puncture test to determine characteristic parameters (breaking force and breaking deformation), and which provide a measurement of the gel rigidity [8].

Cheese ripening is a complex process in which biochemical changes such as moisture loss, change in pH, fat migration, the progressive breakdown of caseins to smaller peptides and the gradual accumulation of free amino acids take place [9]. Cheese ripening continues during the storage, distribution and retail of the product. The degree of residual ripening depends on the cheese manufacturing parameters, as well as the storage temperature and duration. Packaging requirements thus vary according to the cheese type, as the maturation rate, water content and mechanical stability will depend on the composition. Nevertheless, the factors to be considered when selecting a packaging material are almost the same for all cheeses, i.e., permeability to gases, mechanical properties and transparency [10,11,12].

Vacuum packaging and modified atmosphere packaging have acquired great interest in the manufacture of food products, as means of preserving the quality and improving the image of products, and for extending the commercial shelf life and minimising the use of additives. Current research is examining the application of biopolymers incorporated into metal nanoparticles to extend the shelf life of different varieties of cheese [13,14]. Although there is a growing degree of environmental concern about the use of non-biodegradable materials in cheese packaging, synthetic packaging films prevent dehydration, and consequent weight loss and the absorption of undesirable odours from the surrounding environment [15,16]. For optimal packaging, the permeability equilibrates the rate of water loss from cheese, with the flux of moisture permeating out of the package, thus producing water activity that minimises the growth of mould on the surface and simultaneously yields good textural and sensory properties [17].

Sensory analysis enables the determination of the possible difference between the sensory characteristics of vacuum-packed cheeses and unpackaged (natural) cheeses. The absence of oxygen reduces the respiration rate and microbial growth, and it also slows down enzymatic changes. It is important to carry out sensory analysis, because cheese needs to maintain good sensorial characteristics to guarantee its quality, specifically cheese with PDO status. The storage temperature greatly influences the quality of cheese. In addition, many cheeses develop their own microflora during storage, which strongly affects the specific characteristics of each variety.

Several studies have reported a relationship between sensory attributes and the physical–chemical properties of cheese [18]. Storage in modified atmospheres contributes to the accumulation of smoke-derived compounds in SSC cheese (negatively correlated with flavour). Therefore, vacuum packaging is the most useful technique for preserving the sensory quality of this cheese.

Considering the long commercial life of SSC cheese relative to other PDO cheeses, we address the technological interest and the possible commercial advantage of vacuum packaging and the chilled storage of samples at 4 °C and 45–55% relative humidity (RH) for 6 months relative to unpackaged samples stored under the same conditions. Another aim of this research was to investigate the possibility of defining certain mechanical parameters, together with viscoelastic magnitudes measured using dynamic rheometry in relation to the sensory textural and chemical attributes of SSC cheese. Both the mechanical and viscoelastic parameters could provide physical data that could be used by regulatory councils to assess quality control and predict consumer response.

## 2. Material and Methods

### 2.1. Cheese Manufacture

Four batches of SSC cheese were made by a local dairy plant known to manufacture premium-grade products, following the Designation of Origin regulations [19]. The cow’s milk used to make the four batches of cheeses contained 3.67 ± 0.04 g fat/100 mL, 3.12 ± 0.05 g protein/100 mL, and 8.62 ± 0.09 g total solids/100 mL. The mean pH of the milk was 6.5 ± 0.1.

Cheeses were transported to the rheology laboratory at 4 °C according to the FIL-IDF (1995) standard [20]. Each cheese represented one sample.

### 2.2. Packaging, Storage and Sampling of the Cheeses

The cheeses were packaged immediately after the 45-day maturation period and subsequent smoking. Four cheeses from each batch were vacuum-packaged (V) in polyethylene and polyamide food bags (thickness, 90 μm; permeability to O_2_ and water vapour, respectively, 50 and 2.6 cm^3^/m^2^, in 24 h, at 23 °C and 75% relative humidity: manufactured by Alfo, Waltenhofen, Germany). The other half of the samples (four cheeses from each batch) were stored naturally (N) (without packaging) in the same refrigeration chamber under the same conditions of refrigeration and relative humidity as for the V cheeses. For each batch, cheeses were removed at time 0 (i.e., after a 45-day aging period) 2, 4 and 6 months for analysis. One half of each cheese was used for mechanical and rheological measurements. The rind of the other half was discarded, and the cheese was grated and stored in airtight containers at −40 °C until biochemical analysis. The same mechanical, rheological and biochemical tests were conducted for each storage time.

### 2.3. Chemical and Physico-Chemical Analyses

Moisture, fat, protein, ash and NaCl contents, pH, titratable acidity (TA), water activity (a_w_) and fat acidity (FA) were analysed following the methods described by Piñeiro-Lago et al. [21]. Nitrogen fractions, i.e., water-soluble nitrogen (WSN), 12% trichloroacetic acid-soluble nitrogen (TCASN) and 5% phosphotungstic acid-soluble nitrogen (PTASN) were determined according to Tavaria et al. [22]. Casein degradation was studied via SDS-PAGE on 12% polyacrylamide gels. Electrophoresis (220 V) was carried out for 30 min. Gels were soaked in a fixing solution of methanol, water and acetic acid (50:43:7) for 15 min, stained in 0.05% (*w*/*v*) Coomassie blue R-250 in methanol, acetic acid and water for 2 h, and then unstained until the background was clear. Gels were visualised and digitally documented using the Gel Doc XR+ imaging system (Bio-Rad, Hércules, Contra Costa County, CA, USA) before analysis with the Quantity One version 4.6.9 software (Bio-Rad, Hércules, Contra Costa County, CA, USA). All analyses were conducted at least in triplicate.

### 2.4. Mechanical Properties: Puncture Test

Puncture tests were performed using the TAXT2i texture analyser from Stable Micro Systems (Aname, Godalming, UK). The tests were carried out at room temperature (20 °C), on cylindrical cheese samples (diameter 20 mm and height 30 mm), which were punctured with a spherical probe (5 mm diameter) attached to a 50 N cell connected to the cross-head on the texture analyser. Force-deformation curves were derived at 1.0 mm/s cross-head speed, with the load as breaking force (BF) and the depth of depression as breaking deformation (BD) being recorded when the cheese sample lost its strength and broke into pieces. All analyses were conducted at least in sextuplicate.

### 2.5. Small Amplitude Oscillatory Shear (SAOS) Measurements

SAOS tests were performed using a Haake RheoStress RS600 rheometer (Thermo Electron Corp., Karlsruhe, Germany). The measurements were carried out using parallel-plate geometry (20 mm diameter, 1 mm gap). Disc-shaped samples were placed on the bottom plate of the rheometer and were left to rest for 15 min before analysis, to ensure both thermal and mechanical equilibrium at the time of measurement. Samples were covered with a steel-solvent trap to preserve the moisture content during testing. Temperature was controlled to within 0.1 °C using a Peltier element in the lower plate. Quintuplicate SAOS tests were carried out.

### 2.6. Stress Sweep Tests

The linear viscoelastic region (LVER) for SSC cheese was determined by running stress sweeps at a constant angular frequency (6.28 rad/s) and a constant temperature (20 °C). The stress sweeps were conducted by varying the shear stress (σ) of the input signal from 25 to 4000 Pa. A total of 300 points were recorded in continuous mode. Changes in the elastic component (G’), viscous component (G’’), complex modulus—G* = [G’^2^ + G’’^2^]^0.5^—and phase angle (δ)—tanδ = G’’/G’—were recorded. The critical (maximum) values of the amplitude sweeps are stress amplitude (σ_max_) and strain amplitude (γ_max_). To obtain the limit of the LVER, the continuous decrease in G* was determined with increasing σ up to (±10%), previously assessed using creep and recovery tests [23].

### 2.7. Mechanical Spectra

Cheese samples were subjected to stress that varied harmonically over time at variable frequencies. The strain amplitude was set at γ = 0.5% within the LVER, and frequency sweeps were run from 0.63 rad/s to 63 rad/s at 20 °C.

### 2.8. Creep and Recovery Tests

An instantaneous stress σ_0_ in the LVER was applied for 600 s to each sample in the creep tests, and the resulting change in strain over time—γ(t)—was monitored. When the stress was released, some recovery was observed for 600 s. The creep and recovery data were described in terms of the shear compliance function, J(t) = γ(t)/σ_0_ [24].

### 2.9. Sensory Analysis

Sensory analysis of each cheese sample in the four batches was carried out after chilled storage of the cheeses for 2, 4 and 6 months. Trained panellists provided sensory results for the V and N cheeses (N) after each storage time. Thus, the descriptive sensory analysis was performed by a trained sensory panel of 10 people aged between 20 and 50 years old. Each person was presented with two randomly arranged samples: one of V cheese and another of N cheese. Samples were prepared and presented in triangular pieces of thickness 5 cm, which were labelled, ordered and placed in plastic dishes beside a tasting questionnaire. The tasting procedure was carried out using the experimental steps included in the process described by Chamorro [25]: visual examination (external and internal appearance), touch (texture), olfactory sensing (first impression of odour intensity) and oral tasting (aromatic, gustative and tactile characteristics). The intensity of each character was scored as follows: 1 (null or very low), 5 (medium) and 9 (high). The final score was calculated as the arithmetic mean of all data.

Other organoleptic characteristics (preference), such as appearance, odour, texture, acid taste and flavour, were evaluated on a 7-point scale (where 1 = low and 7 = high) [26]. A global score was obtained adding the average ratings for appearance, texture and taste attributes, multiplied by 1, 3, and 6, respectively [27].

### 2.10. Statistical Analysis

For statistical analysis of the biochemical parameters in the V and N samples in relation to storage time, analysis of variance (ANOVA) and post hoc Tukey tests were applied using 8.0 windows Statistical Software (Tulsa, OK, USA). The expanded uncertainty limit (EUL) was calculated to enable the comparison of rheological parameters and mechanical properties. The EUL indicates the experimental uncertainty and is calculated as the mean standard deviation multiplied by the Student t factor corresponding to the specific number of replicates for each test, and maintaining the same significance level (*p* < 0.05).

Biochemical, rheological, and sensory parameters were compared by calculating Pearson correlation coefficients, considering the significance levels of *p* < 0.05, *p* < 0.01 and *p* < 0.001.

## 3. Results and Discussion

### 3.1. Biochemical Parameters during Chilled Storage

The loss of moisture was greater in N cheeses, and the total solids (TS) content was significantly higher (*p* < 0.05) (by 16%) after chilled storage for 6 months. In the V cheeses, the TS remained constant (~65 g/100 g of cheese) (Table 1). Significant differences (*p* < 0.05) between the N and V cheeses were observed after 2 months of storage. This trend has also been observed in other vacuum-packed cheeses, such as Cameros [28], Parmigiano Reggiano [29], Provolone [30] and Arzúa-Ulloa [31].

Water loss during storage is prevented at the surface of vacuum packaged cheese, and as a consequence of the outflow of fat, the rind becomes smoother and more waxy [17,32], and changes occur in the external appearance [29,33].

In the V cheeses, the fat content remained constant (around 55 g/100 g TS), although fat migration was visible during storage. However, there were no significant differences between the N and V cheeses. The mean protein (TN × 6.38), ash and NaCl contents, expressed as percentage of TS, remained almost constant throughout the storage period, irrespective of whether the cheeses were packaged or not, with final mean values of around 36%, 5.5% and 2.4%, respectively.

The final quality of cheese is largely determined by the salt in moisture concentration due to the influence on the development of lactic acid bacteria, enzyme activity and the biochemical relationships during ripening [34]. Values of NaCl/100 g of moisture increased significantly (*p* < 0.05) between 0 and 6 months (by 52%) in N cheeses as a consequence of the dehydration process. The NaCl/moisture ratio was significantly higher (*p* < 0.05) in N than in V samples after 4 and 6 months of storage (Table 1), which naturally increased the ionic strength in the cheese mass.

As a consequence of the greater moisture loss in N cheeses, the M/P ratio decreased significantly (*p* < 0.05) (by 36%) during storage. However, the relationship remained almost constant in V cheeses during storage (Table 1). This finding shows the important role of vacuum packaging in preserving the moisture content of cheeses—and consequently, the concentration of the principal component of the cheese structure (casein matrix)—because the mechanical properties of the cheese network largely depend on the firmness of the casein matrix. The observed trend was maintained from the beginning of storage, as indicated by the significantly higher M/P ratio (*p* < 0.05) in V than in N cheeses for a fixed time of chilled storage.

The pH of the samples remained almost constant (~5.8) during storage, with no significant differences between the N and V cheeses. This can be explained by lactose metabolisation and the formation of lactic acid at the initial stages of the elaboration, and by the moderate level of proteolysis. Although the titratable acidity (TA) did not vary significantly during storage, a gradual increase in the TA values after 6 months was observed in the N and V cheeses.

As a result of the increase in TS in the N cheeses, the water activity (a_w_) in these samples decreased during storage, notably after 4 months. On the other hand, a_w_ remained constant in the V cheeses throughout the entire storage period, which is consistent with the stable moisture contents of these samples.

The changes in the different nitrogen fractions, expressed relative to the total nitrogen (TN) content, during chilled storage, are shown in Table 2, and the SDS-PAGE electrophoretograms corresponding to the different storage times are shown in Figure 1.

Moderate protein degradation occurred during storage and was similar in the V and N cheeses. The WSN and TCASN fractions increased by, respectively, 1.5% and 1.8% during the storage of both V and N cheeses, and the percentages of PTASN increased by 2.

The final values of WSN, TCASN, and PTASN were similar or even lower than those determined in other varieties of cheese at the end of the maturation stage [35,36], and they were even lower than those reported for San Simón da Costa cheese after 45 days of ripening [19,37].

Vacuum packaging did not affect the proteolysis rate, although proteases and peptidases of microbial origin would be expected to have different effects. Under anaerobic conditions, when the amount of oxygen is limited, the microbiota established during the storage of V cheeses will be different from that in N cheeses [26]. This trend has already been demonstrated in other cheeses such as Cameros [28], Parmigiano Reggiano [29], Provolone [30,38], Saloio [17], Crottin de Chavignol [39] and Arzúa-Ulloa [31]. It therefore seems that chilled storage (4 °C) will slow down the enzymatic action, which is consistent with the low intensity of proteolytic reactions observed at this stage.

No significant differences in casein degradation in the N and V cheeses were observed in relation to chilled storage. This enables us to conclude that the rate of proteolysis was similar in both N and V cheeses. The electrophoretic profile indicated that the main agents involved in the casein degradation at this stage were the proteases in the lactic-acid flora present. The quantification of caseins and their degradation products confirmed the results obtained for the classic nitrogen fractions, showing that the SSC cheese underwent moderate proteolysis in both extent and depth throughout storage.

During the storage of the cheeses, the acidity index (AI) increased slightly, particularly in N cheeses (Table 2). After 6 months, the AI values were significantly higher in N than in V cheeses. Several studies on the effects of vacuum packaging on lipolytic phenomena have reported a similar trend to that observed in SSC cheese [28,29,30]. The lower values of AI in V cheeses may be related to the effective loss of fat, which migrates to the exterior due to vacuum application [29]. Moreover, it may also be related to the reduction in enzymatic activity, as the main lipolytic enzyme (LPL) is only active at the fat–water interface [40], which will be reduced because of fat migration.

### 3.2. Changes in Mechanical Properties during Chilled Storage

In N samples, the BF values increased greatly (by 219%) between the months 0 and 6 (Table 3). Specifically, between months 0 and 2, a significant increase in BF (90%) was observed simultaneously, with a significant decrease in BD (29%). This result reflects the hardening undergone by N samples during the first two months of storage at 4 °C, which may be related to the significant loss of moisture (Table 1) and the corresponding decrease in water activity in the N samples (Table 1). The degree of hardening was also indicated by the significant and positive correlation between BF and total solids (TS) (*p* = 0.001, r = 0.86).

Regarding the influence of vacuum packaging on the mechanical characteristics, in the initial 0–2 months of storage, the BF increased significantly (40%), although obviously to a lesser extent than in the N samples. Moreover, BD decreased significantly (19%), also to a lesser extent than in the N samples (Table 3). These findings can be explained by considering that in V cheeses (0–2 months), vacuum packaging may produce some internal pressure that induces fat migration through the cheese network to produce some structural packing [29].

### 3.3. Viscoelastic Properties in the LVER

#### 3.3.1. Changes in LVER for SSC Cheese during Chilled Storage (Stress Sweep)

For the N samples, γ_max_ decreased significantly (*p* < 0.05) over 0–6 months, while σ_max_ increased significantly (*p* < 0.05) over the same period. In addition, G* values also increased in the same way as σ_max_ (Table 4). These data reflected the notable increase in the gel strength between months 0 and 6 (460%), with the high G* [41] occurring due to the loss of moisture during chilled storage, resulting in the strengthening of the casein matrix. In addition, lipolysis was significantly higher (*p* < 0.05) in N samples in 0–6 months (Table 1). Thus, moisture and fat can act as, respectively, plasticiser and filler in the casein matrix [42]. When fat molecules are broken down, the total filler volume can decrease. Thus, the texture of cheese affected by intense lipolysis is harder than that of cheese containing intact fat, which may also reinforce the casein–casein links in the micellar matrix and the subsequent increased firmness of samples.

Similarly, the conformational stability (σ_max_) also increased (130%) throughout the 0–6 months of refrigerated storage (Table 4). These results are consistent with the narrowing of the γ_max_ (62%) in 0–6 months (Table 4). Thus, G* of strongly associated particles (N cheese networks) tends to be highly strain-dependent, with the linear response being confined to a low strain amplitude [43].

However, the phase angles (δ) of the N and V samples remained constant over 0–6 months (Table 4). This indicates that the increase in rigidity was compatible with the same solid-like character in an energy-stable network that retains the elastic character (ideal network fraction) of cheese [44]. This finding also helps to clarify the viscoelastic effects of the moderate casein degradation observed in N cheeses via the measurement of the nitrogen fractions (Table 2) and electrophoresis (Figure 1) during storage.

In the V cheeses, the characteristic LVER parameters (σ_max_, γ_max_ and G*) followed similar trends as in the N samples. Specifically, σ_max_ and G* were significantly lower in V than in N cheeses after 2, 4 and 6 months (Table 4), consistent with the significantly higher moisture contents of V cheeses, which provide a softer casein-matrix.

#### 3.3.2. Frequency Sweep

From mechanical spectra, all cheeses behaved as “true gels, as G’ > G’’ throughout the frequency range, being almost independent of frequency (ω) (Figure 2) [45]. G’ and G’’ were fitted to the angular frequency by the power law (Equations (1) and (2)), respectively, where G_0_’ is the storage modulus and G_0_’’ is the loss modulus at 1 rad/s. The n’ and n’’ exponents provide the frequency dependence (time-stability) of the gel network [46].
(1)$$G^{’}={G_{0}}^{’}\cdot \omega^{n^{’}}$$
(2)$$G^{”}={G_{0}}^{”}\cdot \omega^{n^{”}}$$

In general, the values of both the G’ and G’’ moduli were higher in N than in the V samples over the entire frequency range, irrespective of the storage time. The difference in G’ and G’’ increased gradually, up to 6 months (Figure 2). This finding is also reflected in the values of G_0_’ and G_0_’’, which increased significantly over time (0–6 months) (Table 5). Specifically, the increase in G_0_’ was higher in the N samples (275%) than in the V samples (110%) for the same time period. The G_0_’’ parameter also increased (286%) more in N samples than in V samples (115%) (0–6 months). These results indicate the formation of a more compacted casein matrix at the end of the storage time, which was more pronounced in the N than in the V samples (Table 5). The notable moisture loss in N cheeses reinforces the casein interactions, producing larger casein aggregates with a high molecular weight, linked by physical links (hydrogen bonds, and electrostatic and hydrophobic interactions) [3]. These observations were corroborated by the significant positive correlations between G_0_’ and TS (*p* = 0.002; r = 0.60), and G_0_’’and TS (*p* = 0.001; r = 0.62).

The values of n’ and n’’ were similar and statistically indistinguishable, irrespective of the storage time for N and V samples (Table 4). When the n’ and n’’ exponents have low and similar values, the gel-like character of cheeses will be maintained for longer (lower frequencies). This result reveals some structural permanency that is associated with the temporal stability of the gel network.

#### 3.3.3. Creep–Recovery Compliance

In general, when the load was applied at time t = 0, instantaneous (vertical) deformation (initial elasticity) was observed. All samples exhibited a non-linear response to instantaneous stress (viscoelasticity). The samples thus exhibited permanent deformation during the recovery stage due to the viscous component which is related to the non-ideal network fraction [44].

The values of creep-recovery compliance during storage were lower in N than in V cheeses. Specifically, the lowest J(t) values for the N sample during creep and recovery times was observed at 6 months. Moreover, the lower permanent deformation, i.e., J(t) during the recovery process, indicates less structural damage during the loading time, and less irreversible breakage of the inter-casein and matrix–filler associations [47]. Conversely, the highest J(t) values for both the creep and recovery stages were observed at t = 0 (Figure 3).

These data can be quantified in terms of the percentage elasticity (Equation (3)).
(3)$$\%\;elasticity=\left(\frac{J_{max}-J_{min}}{J_{max}}\right)\cdot 100$$
where *J_max_* represents the value of creep compliance at t = 600 s (creep curve), and *J_min_* represents the value of the recovery compliance at 1200 s (recovery curve) [48].

The lowest percentage of elasticity was observed at t = 0 (Table 6). The elasticity then increased slightly during 0–2 months in both the N and V cheeses. It then remained almost constant in both the N and V samples during the subsequent 2–6 months (Table 6). This finding demonstrates the specific reticular stability of the SSC cheese network during storage, irrespective of the conservation method, in line with the stable phase angles during chilled storage, obtained from the stress sweep tests.

Moreover, the J(t) data from the creep values are used to calculate the relaxation modulus G(t). Thus, log J(t) vs. log t divided by the creep time, has a slope m << 1 (data not shown), and G(t) then becomes the reciprocal of J(t) [48]. From the power law fit of G(t) *vs*. t, another measurement of the gel strength (*S*) can be obtained, together with the relaxation exponent (*n*) (Equation (4)):(4)$$G(t)=S\cdot t^{-n}$$
where *S* (Pa·s^n^) provides a measure of the reticular firmness based on the number and the strength of junction zones between casein domains (nodules). The *n* parameter is the relaxation exponent [44] and is related to the density of junctions, i.e., the degree of connectivity in the gel network [49]. Thus, n provides the degree of temporal dependency (level of permanency) of the cross links in the cheese network at higher time scales than n’ and n’’ (mechanical spectra) [48]. In general, low values of the n exponent indicate a lower time dependence of the cross links, resulting in a more time-stable, and hence more permanent gel network. The specific stability may be a consequence of the higher energy content of different bonds, such as hydrogen bonding, ionic, dipolar and hydrophobic interactions, which together lengthen the lifetime of bonds in junctions [50].

For N samples in the 0–6 month interval, the S parameter increased significantly (470%) (Table 6). This finding is consistent with the greater loss of moisture content in N cheeses, which was more pronounced during the final period of storage. Thus, the concentration of the casein matrix was enhanced in the cheese network as the storage time increased, resulting in a greater cohesive strength in the continuous phase, associated with more numerous and extended junction zones in the gel network [51]. This is characteristically observed as cheese ages [52]. During aging, after the network has been completely developed, the gel strength increases due to an increase in thickness at the junctions [53]. This finding is reinforced by the presence of degradation products such as free amino acids, as deduced from the higher content of PTASN in N cheeses, which was more notable after 6 months of storage (Table 2). These smaller structural units (free amino acids) have a net electrostatic charge, and they can link water molecules and may act as surfactants between the fat phase and the casein matrix, thus reinforcing the contact zone in the fat–casein inter-phase, resulting in a coupled, firmer cheese network due to the greater thickness in junction zones after 6 months of storage.

In the V samples, gel strength also increased with storage time, but with a somewhat lower percentage of elasticity than in the N samples. Thus, for V cheeses, the S parameter increased (102%) during the 0–6-month storage period (Table 6), in accordance with the trend observed for viscoelastic moduli from SAOS tests. The greater number of water molecules and the lower proportion of free amino acids possibly indicates that a large proportion of these amino acids pass to the aqueous phase in the cheese network, with the surfactant role between the fat filler and the casein matrix being less important in the V samples.

Moreover, part of the effect of packaging on the casein matrix in V samples (0–6 months) can be explained by the pressure exerted by the vacuum packaging and its influence on fat mobility in the cheese network. Thus, fat may be filtered in the cheese structure; moreover, the filtered fat may homogenise the cheese network in V samples during the storage period, thus improving the aggregation of casein particles and consequently reinforcing the solid matrix of the cheese network. This specific structural reinforcement has a particular benefit, indicated by the increase in deformability, as previously determined in stress amplitudes (stress sweep tests). This idea is consistent with the lower gel strength values in V than in N cheeses for a fixed storage time (Table 6).

On the other hand, in both N and V samples, the value of the relaxation exponent n was constant during the 0–6 month storage period (Table 6). This finding indicates the temporal equilibrium in the casein matrix, which is consistent with the structural stability reflected in the constant values of the percentage of elasticity (Table 6). Thus, the nature (lifetime, i.e., the strength of inter-casein bonds) did not change during the storage period.

### 3.4. Sensory Analysis

The results obtained in the evaluation of the oral tactile phase for N and V cheeses after 2, 4 and 6 months of storage at 4 °C are shown in Figure 4. The results are given as the average total scores for each attribute and sample.

The tactile characters of cheeses changed during the storage time. Specifically, the firmness increased, with significant differences between the N (mean value, 7) and V (mean value, 3) cheeses at six months (Figure 4c). The increase in the sensory firmness is expected due to moisture loss (Table 1), which leads to a harder structure.

Regarding the friability, i.e., the ability of the sample to produce numerous portions from the start of chewing, values remained almost constant over 2–4 months in both the N and V samples (Figure 4). However, the friability of the N samples was higher after 6 months (average value = 6), and significant differences were observed between the N and V cheeses (Figure 4c). A similar trend was also observed in the grain and perception of the microstructure.

Viscosity is the capacity of a sample to deform after pressing during chewing. Differences in viscosity between the N and V cheeses were observed from 4 months (Figure 4b), and it was enhanced at 6 months, reaching the highest values in the V samples (Figure 4c). This finding confirms that vacuum packaging preserves juiciness and tenderness during long-term storage.

Adherence is the degree to which the chewed sample adheres to the surface of mouth and teeth [54]. The adherence increased in the V samples during the 6 months (Figure 4). The sensory humidity (oral perception of the degree of humidity) was different in the N and V cheeses from 4 months onwards (Figure 4), and the difference was greatest at 6 months.

For chewiness at 4 months, some differences were observed between the N and V cheeses, which were greater at 6 months of storage. The chewiness (mean value, 6) of the N samples increased to 7 after 6 months (Figure 4c).

The gumminess (the ability of cheese, after being bitten, to deform and extend before breaking) and fatty character differed between the N and V samples at 6 months (Figure 4a–c). These differences were stabilised via the dehydration of the N cheeses during chilled storage. In conclusion, the N cheese can be characterised by its firmness/hardness, and the V cheese by its viscosity, adhesiveness, and sensory humidity.

The organoleptic characteristics (preference) of the N and V samples of SSC cheeses were also evaluated after 2, 4, and 6 months of chilled storage (Figure 5). After 2 months, no differences were observed in these characteristics between the N and V cheeses (Figure 5a). After 4 months, flavour was scored lower in V than in N cheeses, because of the greater perception of an acid flavour, which was considered as being undesirable by the panellists (Figure 5b).

The difference in flavour between the N and V cheeses can be explained by the higher moisture content of the V samples, together with the effect of packaging, which retains the peculiar smoky aroma.

Although no correlation between nitrogen fractions and flavour was observed in the SSC cheese, this has also been observed in other types of cheese. Thus, Ballesteros et al. [55] reported a direct relationship between ASN, FSN and flavour in Manchego cheese. On the other hand, Mendía et al. [56] reported higher values of olfactory and gustatory attributes for Idiazábal cheese, which exhibited intense α, β-casein degradation. However, Madkor et al. [57] and Hayaloglu et al. [58] proposed a direct relationship between flavour intensity and the concentration of the nitrogen fractions and free amino acids in cheeses.

After storage of the samples for 6 months, differences between the N and V cheeses were observed in four of the five attributes, with the lowest values corresponding to the V samples (Figure 5c): texture (soft), acid flavour (intense), strong odour (smoke) and flavour. The acid flavour was significantly (*p* < 0.05) stronger in the N than V cheeses, and the changes (N sample) may be related to the increase in the concentration of the free fatty acids, particularly short chain fatty acids, during storage [59,60,61]. The free fatty acids are transformed into keto acids via β-oxidation reactions. The keto acids are then decarboxylated and transformed into methyl-ketones, which in turn are reduced to alcohols.

Because of the higher rate of lipolysis in N cheeses during storage, some increase in the volatile compounds that develop the cheese aroma will occur. Garabal et al. [26] reported that high concentrations of various alcohols and compounds derived from smoke had a negative influence on the flavour of unpackaged SSC cheese, producing an intense smoky aroma. Thus, as was observed for other cheeses, where the increase in the intensity of the aroma is produced by different volatile or non-volatile components and their proportions, which have been produced during proteolysis and lipolysis [61]. On the other hand, aroma scores may be related to the flavour values, and the same trend may be observed. This relationship was also observed in Manchego cheese [55].

In the SSC cheese, the buttery and smoky aromas (not excessive), which are positive characteristics according to San Simón da Costa PDO [1], were well accepted by the panellists. The difference in the salty taste of the N and V cheeses is associated with an increase in the salt concentration due to the moisture loss during storage. This change in the ionic strength promotes a change in the mouthfeel of the salt diffusion during chewing [62].

The rind was smooth, greasy and bright due to the fat migration in the V cheeses after 4 and 6 months. This was a negative characteristic according to the panellists. The dehydration of the N cheeses was observed from 4 months onwards, which explains the increased thickness of the rind. This effect was more pronounced after 6 months, and it was negatively evaluated by the panellists. N samples at 6 months scored higher on acid flavour, odour and flavour (Figure 5c).

The average scores for the appearance, texture and flavour attributes were multiplied by, respectively, 1, 3 and 6, and were added together to produce a global score [27]. The global score was lower for the V than for the N cheese, and the difference was greater and significant after 6 months of storage, reaching values of 45 ± 1 for the V cheeses and 51 ± 4 for the N cheeses. The present data are consistent with those obtained by Garabal et al. [26], who reported a more favourable sensory evaluation of cheese stored without packaging after 45 days of ripening, than of the corresponding packaged cheese.

Although the texture of the V cheeses was smoother and more elastic, similar to the “semi-elastic” descriptor proposed by the PDO, the panellists preferred the N cheeses. Fat migration in the V cheeses may be related to changes leading to a more friable, elastic structure [29].

### 3.5. Correlation between Sensory and Rheological Parameters

Correlations between rheological (viscoelastic and mechanical) parameters and the analogous sensory parameters were established. Thus, sensory firmness was significantly and positively correlated with the rheological parameters associated with physical firmness, such as BF, G*, G_0_’, G_0_’’ and σ_max_ (Table 7). In addition, friability was significantly and positively correlated with BF, G*, G_0_’, G_0_’’ and σ_max_. Thus, the moisture loss caused some hardening of the cheese network, and the number of portions on initial chewing will be greater, which would explain the significant negative correlations between the friability and creep compliance (J_0_ and J_f_) (Table 7). Both values indicate a temporal deformability, and thus, the negative sign is reasonable because the friability is similar to the sensory firmness.

Sensory humidity and viscosity were negatively correlated with the rheological parameters related to the physical firmness, such as G*, G_0_’, G_0_’’, σ_max_ and BF, and positively correlated with compliance factors (J_0_ and J_f_) (Table 7). Thus, viscosity and sensory moisture reflect the lubrication capacity of water in the cheese network, producing the tenderness and juiciness of samples. These characteristics are related to a softer and more deformable cheese network (lower viscoelastic moduli and higher compliance).

Gumminess was positively correlated with compliance factors (J_0_ and J_f_). Thus, the cheese deformability and extensibility before breaking was positively correlated with a higher temporal deformability (high compliance values). Therefore, a more time-deformable structure needs a greater amount of energy to disintegrate a piece of cheese to a state that is ready for swallowing. Conversely, gumminess was negatively correlated with rheological parameters related to physical firmness over a short time-scale (BF, G_0_’ and G_0_’’). However, on a longer time scale, gel strength (S) was significantly and positively correlated with gumminess, which is consistent with the greater amount of energy required to crumble the sample prior to swallowing.

The correlations between tactile phase attributes (i.e., sensory firmness, friability, grainy, viscosity and sensory humidity) and the rheological, mechanical and viscoelastic parameters (such as BF, G*, S, G_0_’, G_0_’’, J_f_, and J_0_) enable the proposal of an experimental basis for the relationship between physical data (instrumental texture) and sensorial attributes (sensory texture).

## 4. Conclusions

Increased gel strength is the principal rheological change observed during the chilled storage of SCC cheese. It is more intense in unpackaged samples because of the higher moisture loss, and this is due to an increased density of cross-links in the protein matrix, an increased ionic strength caused by the increased salt/water ratio, and the increase in free amino acids and low molecular weight peptides. Both natural and vacuum-packaged cheeses exhibited a high level of temporal and bond-energy stability during the storage time. The physical measurement of this parameter would serve as a useful tool for characterising the San Simón da Costa cheeses.

Unpackaged stored samples were awarded the best sensorial evaluations by the panel of tasters. The mouthfeel attributes of firmness, friability and gumminess, and the perception of the microstructure and the mechanical parameters (breaking force) were favourably evaluated by the tasting panel. The correlations between rheological (puncture tests and SAOS data) and sensory attributes (sensory firmness, friability, graininess, viscosity and sensory humidity) provide an experimental basis for the relationship between the instrumental (physical data) and sensorial textures. Thus, the sensory attributes of the San Simón da Costa cheese can be determined using the set of rheological parameters established in this study. Nevertheless, further research should be carried out to determine the impacts of new biomaterials used in cheese coatings and packaging.

## Figures and Tables

**Figure 1 foods-12-01381-f001:**
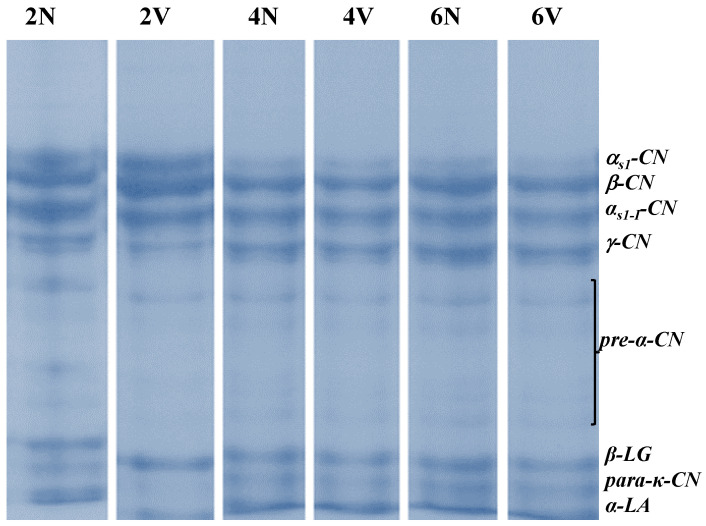
SDS-PAGE electrophoretograms of caseins and their degradation products during chilled storage of natural (N) and vacuum-packaged (V) samples of San Simón da Costa cheese.

**Figure 2 foods-12-01381-f002:**
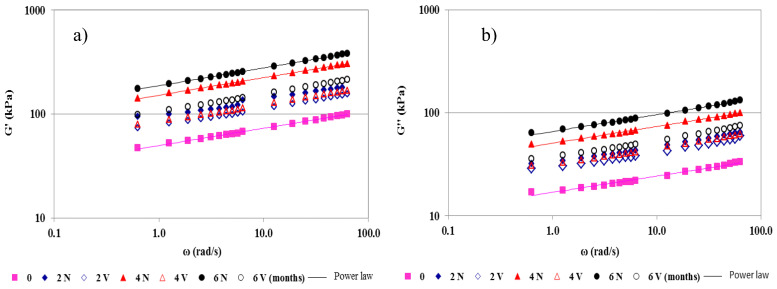
Mechanical spectra of natural (N) and vacuum-packaged (V) San Simón da Costa cheese during chilled storage. Storage modulus (G’) (**a**) and loss modulus (G’’) (**b**), T = 20 °C.

**Figure 3 foods-12-01381-f003:**
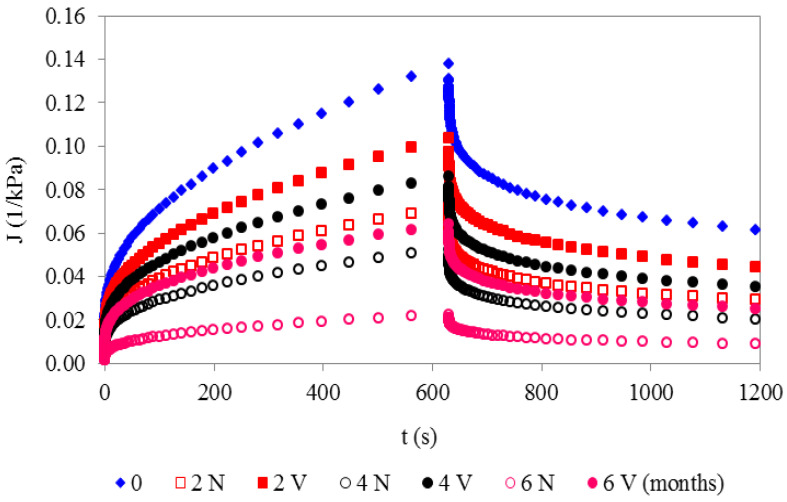
Creep–recovery compliance of natural (N) and vacuum-packaged (V) San Simón da Costa cheese during chilled storage, T = 20 °C.

**Figure 4 foods-12-01381-f004:**
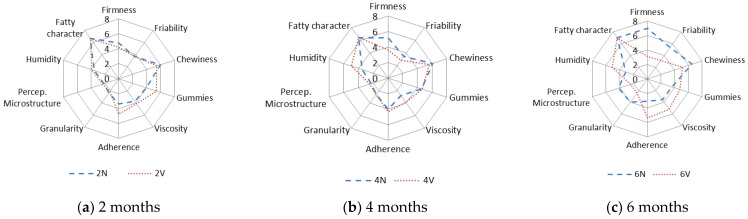
Characteristics of oral tactile phase of natural (N) (**---**) and vacuum-packaged (V) (**…**) San Simón da Costa cheese during chilled storage.

**Figure 5 foods-12-01381-f005:**
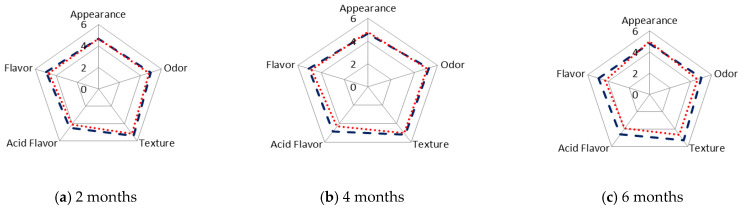
Organoleptic characteristics of natural (N) (**---**) and vacuum-packaged (V) (**…**) San Simón da Costa cheese during chilled storage.

**Table 1 foods-12-01381-t001:** Compositional and physico-chemical parameters during chilled storage of natural (without packaging) and vacuum-packaged San Simón da Costa cheese.

Sample	Parameter	Storage Time (Months)			
		0	2	4	6
Natural cheese	TS ^1^	62.9 ± 1.5 ^a^	68.5 ± 2.2 ^b,^*	70.6 ± 1.2 ^b,c,^*	73.0 ± 1.4 ^c,^*
	Fat ^2^	53.8 ± 1.5 ^a^	53.1 ± 2.5 ^a^	56.2 ± 3.2 ^a^	56.7 ± 3.6 ^a^
	Protein ^2^	36.5 ± 1.1 ^a^	35.4 ± 1.1 ^a^	35.25 ± 0.78 ^a^	35.96 ± 0.97 ^a^
	Ash ^2^	5.72 ± 0.19 ^a^	5.64 ± 0.16 ^a^	5.57 ± 0.28 ^a^	5.53 ± 0.25 ^a^
	NaCl ^2^	2.53 ± 0.26 ^a^	2.23 ± 0.06 ^a^	2.39 ± 0.07 ^a^	2.39 ± 0.25 ^a^
	NaCl ^3^	4.28 ± 0.56 ^a^	4.75 ± 0.52 ^a^	5.80 ± 0.35 ^a,b,^*	6.50 ± 0.86 ^b,^*
	M/P	1.62 ± 0.09 ^a^	1.31 ± 0.11 ^b,^*	1.18 ± 0.06 ^b,^*	1.04 ± 0.07 ^c,^*
	pH	5.73 ± 0.07 ^a^	5.73 ± 0.05 ^a^	5.81 ± 0.13 ^a^	5.77 ± 0.10 ^a^
	TA^4^	1.28 ± 0.28 ^a^	1.43 ± 0.18 ^a^	1.56 ± 0.17 ^a^	1.70 ± 0.40 ^a^
	a_w_	0.960 ± 0.003 ^a^	0.958 ± 0.011 ^a^	0.941 ± 0.006 ^a,b^	0.932 ± 0.004 ^b,^*
Vacuum-packaged cheese	TS		65.3 ± 1.9 ^a^	65.0 ± 1.6 ^a^	65.4 ± 1.3 ^a^
	Fat ^2^		55.45 ± 0.92 ^a^	54.4 ± 2.2 ^a^	54.6 ± 2.1 ^a^
	Protein ^2^		35.4 ± 1.0 ^a^	35.8 ± 1.0 ^a^	36.3 ± 1.2 ^a^
	Ash ^2^		5.45 ± 0.38 ^a^	5.69 ± 0.21 ^a^	5.66 ± 0.30 ^a^
	NaCl ^2^		2.26 ± 0.17 ^a^	2.37 ± 0.19 ^a^	2.42 ± 0.12 ^a^
	NaCl ^3^		4.22 ± 0.40 ^a^	4.38 ± 0.45 ^a^	4.53 ± 0.37 ^a^
	M/P		1.51 ± 0.09 ^a^	1.50 ± 0.08 ^a^	1.46 ± 0.11 ^a^
	pH		5.76 ± 0.06 ^a^	5.83 ± 0.10 ^a^	5.73 ± 0.08 ^a^
	TA ^4^		1.52 ± 0.21 ^a^	1.57 ± 0.30 ^a^	1.74 ± 0.40 ^a^
	a_w_		0.962 ± 0.005 ^a^	0.958 ± 0.006 ^a^	0.955 ± 0.006 ^a^

^1^ g/100 g cheese; ^2^ g/100 g TS; ^3^ g/100 g moisture; M/P: Moisture/Protein. ^4^ g lactic acid/100 g TS. TS: total solids; TA: titratable acidity; a_w_: water activity. ^a–c^ Different letters in the same row indicate significant differences (*p* < 0.05) for different storage times. * in the same column indicates significant differences (*p* < 0.05) between natural and vacuum-packaged cheeses for each parameter at fixed storage time.

**Table 2 foods-12-01381-t002:** Changes in the nitrogen fractions (expressed as g/100 g of total nitrogen) and acidity index (AI) (expressed as mg KOH/g fat) during chilled storage of natural (N) and vacuum-packaged (V) San Simón da Costa cheese.

Samples	Parameter	Storage Time (Months)			
		0	2	4	6
Natural cheese	WSN	15.3 ± 1.1 ^a^	18.6 ± 1.5 ^b^	21.1 ± 1.6 ^c^	22.7 ± 2.1 ^c^
	TCASN	6.86 ± 0.46 ^a^	10.8 ± 1.6 ^b^	10.76 ± 0.98 ^b^	12.22 ± 0.94 ^b^
	PTASN	0.97 ± 0.22 ^a^	1.35 ± 0.37 ^b^	1.63 ± 0.34 ^b,c^	2.15 ± 0.25 ^c^
	AI	0.12 ± 0.02 ^a^	0.33 ± 0.02 ^b,^*	0.33 ± 0.01 ^b,^*	0.33 ± 0.01 ^b,^*
Vacuum-packaged cheese	WSN		19.4 ± 1.3 ^b^	20.7 ± 1.8 ^b^	23.3 ± 1.5 ^c^
	TCASN		9.71 ± 0.71 ^b^	11.40 ± 0.84 ^c^	12.14 ± 0.68 ^c^
	PTASN		1.42 ± 0.40 ^ab^	1.78 ± 0.64 ^ab^	1.95 ± 0.41 ^b^
	AI		0.21 ± 0.01 ^b^	0.23 ± 0.02 ^c^	0.23 ± 0.02 ^c^

^a–c^ Different letters in the same row indicate significant differences (*p* < 0.05) for different storage times. * in the same column indicates significant differences (*p* < 0.05) between natural (without packaging) and vacuum-packaged cheeses for each parameter at fixed time.

**Table 3 foods-12-01381-t003:** Data for puncture tests: breaking force (BF) and breaking deformation (BD) during chilled storage of natural (N) and vacuum-packaged (V) San Simón da Costa cheese. Data are expressed as mean values ± expanded uncertainty limit, T = 20 °C.

Samples	Parameter	Storage Time (Months)			
		0	2	4	6
Natural cheese	BF (N)	3.95 ± 0.47 ^a^	7.55 ± 0.93 ^b,^*	8.8 ± 1.3 ^b,^*	12.6 ± 1.8 ^c,^*
	BD (mm)	9.14 ± 0.44 ^a^	6.50 ± 0.36 ^b,^*	5.98 ± 0.49 ^b^	6.07 ± 0.51 ^b^
Vacuum-packaged cheese	BF (N)		5.59 ± 0.71 ^b^	5.67 ± 0.62 ^b^	5.60 ± 0.78 ^b^
	BD (mm)		7.41 ± 0.40 ^b^	6.79 ± 0.47 ^b^	6.72 ± 0.34 ^b^

^a–c^ Different letters in the same row indicate significant differences (*p* < 0.05) for different storage times. * in the same column indicates significant differences (*p* < 0.05) between natural (N) and vacuum-packed (V) cheeses for each mechanical property at a fixed storage time.

**Table 4 foods-12-01381-t004:** Comparison of the stress amplitude (σ_max_), strain amplitude (γ_max_), complex modulus (G*) and phase (δ) values during chilled storage of natural (N) and vacuum-packaged (V) San Simón da Costa cheese. Data are expressed as mean values ± expanded uncertainty limit, T = 20 °C.

Samples	Parameter	Storage Time (Months)			
		0	2	4	6
Natural cheese	σ_max_ (Pa)	328 ± 33 ^a^	493 ± 49 ^b,^*	669 ± 67 ^c,^*	808 ± 81 ^d,^*
	γ_max_ (%)	0.62 ± 0.11 ^a^	0.47 ± 0.06 ^b^	0.43 ± 0.06 ^b^	0.25 ± 0.03 ^c,^*
	G* (kPa)	60 ± 10 ^a^	113 ± 18 ^b^	177 ± 28 ^c^	338 ± 39 ^d,^*
	δ (°)	18.01 ± 0.31 ^a^	18.14 ± 0.27 ^a^	18.17 ± 0.26 ^a^	18.19 ± 0.38 ^a^
Vacuum-packaged cheese	σ_max_ (Pa)		402 ± 40 ^b^	493 ± 49 ^b,c^	546 ± 55 ^c^
	γ_max_ (%)		0.46 ± 0.07 ^a,b^	0.44 ± 0.09 ^a,b^	0.38 ± 0.07 ^b^
	G* (kPa)		97 ± 14 ^b^	132 ± 20 ^c^	163 ± 27 ^c^
	δ (°)		18.29 ± 0.24 ^a^	18.24 ± 0.36 ^a^	18.03 ± 0.39 ^a^

^a–d^ Different letters in the same row indicate significant differences (*p* < 0.05) for different storage times. * in the same column indicates significant differences (*p* < 0.05) between natural and vacuum-packaged cheeses for each viscoelastic parameter at a fixed storage time.

**Table 5 foods-12-01381-t005:** Comparison of power law fit parameters (equations 1 and 2) during chilled storage of natural (N) and vacuum-packaged (V) San Simón da Costa cheese. Data are expressed as mean values ± standard deviation of fit parameters. T = 20 °C.

Samples	Parameter	Storage Time (Months)			
		0	2	4	6
Natural cheese	G_0_’ (kPa)	49.78 ± 0.24	89.59 ± 0.36 *	151.98 ± 0.42 *	186.82 ± 0.66 *
	n’	0.165 ± 0.004	0.166 ± 0.004	0.168 ± 0.003	0.174 ± 0.003
	r^2^	0.997	0.998	0.999	0.999
	G_0_’’ (kPa)	17.11 ± 0.13	32.86 ± 0.23 *	50.76 ± 0.33 *	65.98 ± 0.44 *
	n’’	0.172 ± 0.007	0.159 ± 0.006	0.161 ± 0.005	0.167 ± 0.005
	r^2^	0.993	0.994	0.995	0.996
Vacuum-packaged cheese	G_0_’ (kPa)		77.09 ± 0.40	84.36 ± 0.31	104.95 ± 0.37
	n’		0.166 ± 0.004	0.166 ± 0.003	0.172 ± 0.003
	r^2^		0.997	0.998	0.999
	G_0_’’ (kPa)		29.02 ± 0.31	31.60 ± 0.25	36.74 ± 0.28
	n’’		0.161 ± 0.009	0.159 ± 0.007	0.169 ± 0.006
	r^2^		0.987	0.992	0.995

* in the same column indicates significant differences (*p* < 0.05) between natural (N) and vacuum-packaged (V) cheeses for each viscoelastic parameter at a fixed storage time.

**Table 6 foods-12-01381-t006:** Effect of storage time of natural (N) and vacuum-packaged (V) San Simón da Costa cheese on the fitting parameters (Equation (4)) and percentage of elasticity (Equation (3)). Data are expressed as mean values ± standard deviation of fit parameters, T = 20 °C.

Samples	Creep Parameter	Storage Time (Months)			
		0	2	4	6
Natural cheese	S (kPa·s^n^)	44.69 ± 0.40	80.67 ± 0.63 *	114.60 ± 0.84 *	256 ± 13 *
	n	0.24 ± 0.01	0.23 ± 0.01	0.23 ± 0.01	0.24 ± 0.01
	r^2^	0.988	0.990	0.992	0.992
	Elasticity (%)	56	60	61	61
Vacuum-packaged cheese	S (kPa s^n^)		62.29 ± 0.53	70.08 ± 0.51	90.30 ± 0.65
	n		0.24 ± 0.01	0.24 ± 0.01	0.23 ± 0.01
	r^2^		0.989	0.991	0.993
	Elasticity (%)		58	59	61

* in the same column indicates significant differences (*p* < 0.05) between natural (N) and vacuum-packaged (V) cheeses for each parameter at fixed storage time.

**Table 7 foods-12-01381-t007:** Significant correlations between sensory parameters (oral tactile phase) and rheological parameters for natural (N) and vacuum-packaged (V) San Simón da Costa cheeses.

Variable	r	Variable	r
Firmness—BF	0.77 ***	Gumminess—G_0_’	−0.68 ***
Firmness—G_0_’	0.58 ***	Gumminess—G_0_’’	−0.70 ***
Firmness—G_0_’’	0.59 ***	Gumminess—S	0.46 **
Firmness—S	0.73 ***	Gumminess—J_0_	0.37 **
Firmness—G*	0.81 ***	Gumminess—J_f_	0.37 **
Firmness—σ_max_	0.64 ***	Viscosity—BF	−0.49 **
Firmness—J_0_	−0.65 ***	Viscosity—G_0_’	−0.56 ***
Firmness—J_f_	−0.63 ***	Viscosity—G_0_’’	−0.55 ***
Friability—BF	0.65 ***	Viscosity—J_0_	0.33 *
Friability—G_0_’	0.62 ***	Viscosity—J_f_	0.34 *
Friability—G_0_’’	0.65 ***	Moisture—BF	−0.76 ***
Friability—G*	0.69 ***	Moisture—G_0_’	−0.45 **
Friability—σ_max_	0.46 **	Moisture—G_0_’’	−0.47 **
Friability—S	0.80 ***	Moisture—σ_max_	−0.47 **
Friability—J_0_	−0.62 ***	Moisture—G*	−0.44 **
Friability—J_f_	−0.60 ***	Moisture—S	−0.57 ***
Adherence—BF	−0.33 *	Moisture—J_0_	0.60 ***
Gumminess—BF	−0.57 ***	Moisture—J_f_	0.58 ***

* Significant correlation at *p* < 0.05, ** *p* < 0.01, *** *p* < 0.001.

## Data Availability

The data presented in this study are available upon request from the corresponding author.

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
