# Peer review of "Effect of Vacuum Packaging on the Biochemical, Viscoelastic, and Sensory Properties of a Spanish Cheese during Chilled Storage"

_foods, 2023, doi:10.3390/foods12071381_

Round 1

Reviewer 1 Report

The manuscript entitled: "Effect of vacuum packing in the biochemical, viscoelastic and sensory properties of a Spanish cheese during chilled storage" is about the effects of packaging on the rheological, physicochemical, and sensory properties of a certain type of cheese.  It looks interesting to the readers and needs improvement, especially in terms of presentation. The research design is acceptable and the objectives of the research aligned with the journal's aims and scopes. The English language is not smooth and it should check by a native English service/person.

1- Abstract: The abstract needs to rewrite again, starting with the aims of the study, then treatments and methodology, followed by the main results with some quantitative data, and close the abstract with a general conclusion.

2- The introduction part needs also to rewrite. The problem statement is not clear, novelty statement also needs to be clear in this part. Make it short and informative. Some interesting articles that may improve the literature part is: Jafarzadeh, Shima, et al. "Cheese packaging by edible coatings and biodegradable nanocomposites; improvement in shelf life, physicochemical and sensory properties." Trends in Food Science & Technology 116 (2021): 218-231.

3- Materials: Add a subheading for materials.

4- Methodology: Cheese preparation and packaging should be in a separate subheading with complete details of processing. Clearly name your treatments in a table.

5- Methods: Make brief as possible and refer to the references from either a standard or a published article.

6- Results and discussions are enough for the article, recommend shortening the correlation part between sensory and rheological properties.

7- The number of tables and figures is too high for a research article, consider removing some tables or merging them.

Author Response

Authors want to thank both referees for review. In our opinion, their comments, suggestions, and criticisms really improved the manuscript. According to the editor’s recommendation, a revision to the original version was performed. This was mainly accomplished taking into account the reviewer’s reports.

1.- The manuscript entitled: "Effect of vacuum packing in the biochemical, viscoelastic and sensory properties of a Spanish cheese during chilled storage" is about the effects of packaging on the rheological, physicochemical, and sensory properties of a certain type of cheese.  It looks interesting to the readers and needs improvement, especially in terms of presentation. The research design is acceptable and the objectives of the research aligned with the journal's aims and scopes. The English language is not smooth and it should check by a native English service/person.

R: Thank you very much for your positive comments. An expert professional linguistic has checked the English.

2.- Abstract: The abstract needs to rewrite again, starting with the aims of the study, then treatments and methodology, followed by the main results with some quantitative data, and close the abstract with a general conclusion.

R1: Thank you very much. Abstract has been rewritten as the Reviewer suggested.

3.- The introduction part needs also to rewrite. The problem statement is not clear, novelty statement also needs to be clear in this part. Make it short and informative. Some interesting articles that may improve the literature part is: Jafarzadeh, Shima, et al. "Cheese packaging by edible coatings and biodegradable nanocomposites; improvement in shelf life, physicochemical and sensory properties." Trends in Food Science & Technology 116 (2021): 218-231.

R2: Thank you very much. The introduction has been changed to clarify the problem and novelty statement. Thank you very much for the suggested paper. We have introduced the reference you recommend and another article with a new revision. Line 72-77.

  • Jafarzadeh, S.; Salehabadi, A.; Nafchi, A.M.; Oladzadabbasabadi, N.; Jafari, S.M. Cheese packaging by edible coatings and biodegradable nanocomposites; improvement in shelf life, physicochemical and sensory properties. Trends Food Sci. Technol., 2021, 116, 218-231,
  • Paidari, S.; Ahari, H.; Pasqualone, A.; Anvar, A.; Allah Yari Beyk, S.; Moradi, S. Bio-nanocomposites and their potential applications in physiochemical properties of cheese: an updated review. Food Meas. Charact., 2023, in press.

4.- Materials: Add a subheading for materials.

R3: Thank you very much. A clarifying subheading has been added to the materials and methods section.

5.- Methodology: Cheese preparation and packaging should be in a separate subheading with complete details of processing. Clearly name your treatments in a table.

R4: Thank you very much. Cheese preparation and packaging were included in a separate subheading as the Reviewer suggested. Considering your suggestion (number 7) together with those of the Reviewers 2 and 3 to reduce the number of Tables, we think do not include new tables to simplify the manuscript. All the treatments performed have been clarified in the text (section 2.2 of Material and Methods: Packaging, storage, and sampling of the cheeses). Line 105-116.

6.- Methods: Make brief as possible and refer to the references from either a standard or a published article.

R5: Thank you very much. The methods have been simplified and they have been referenced including a previous papers. Line 126-128.

  • Piñeiro-Lago, L.; Franco, I.; Tovar, C.A. Changes in thermoviscoelastic and biochemical properties of Atroncau blancu and roxu Afuega'l Pitu cheese (PDO) during ripening. Food Res. Int., 2020, 137, 109693.

7.- Results and discussions are enough for the article, recommend shortening the correlation part between sensory and rheological properties.

R6: Thank you very much. The correlation between sensory and rheological properties has been reduced as the Reviewer suggested and three tables were removed (table 7, 8 and 9).

8.- The number of tables and figures is too high for a research article, consider removing some tables or merging them.

R7: Thank you very much. The number of tables has been reduced as the Reviewer suggested. Tables 7, 8 and 9 were removed and the results discussed in the related section.

Reviewer 2 Report

1. The manuscript was generally good, there are some grammatical and syntax errors in the text. Thus, the language of paper has to be checked carefully again.

2. Abbreviations used in whole manuscript have to be defined firstly and then their abbreviations have to be used.

3. The abstract of the MS is not well documented to represent the whole manuscript objectives. Thus, it should be rewritten again in order to represent the paper sufficiently.

4. Regarding to the originality of the manuscript in the last paragraph (Introduction), it should be indicated clearly how this manuscript contributes to the existing knowledge. Please also identify and describe the originality of the study described in this manuscript and how it will likely to contribute to the state-of-the-art. Also, The introduction should be enrich by update references in the field of dairy products.

5.Conclusion part is short, should be revised. It should contain main results and suggestions for further or future studies.

6. What about the microbiological analysis for the cheese during storage. I advise to  determine the microbiological load for the samples during storage that related also to the ripening of cheese.

7. Statistical analysis – the number of trials and repetitions is not mentioned; confidential interval not mentioned; test used for differences between samples – not mentioned.

8.  the sensory evaluation should be done by expert persons and how many person was evaluated the sensory of cheese?

Author Response

Authors want to thank both referees for review. In our opinion, their comments, suggestions, and criticisms really improved the manuscript. According to the editor’s recommendation, a revision to the original version was performed. This was mainly accomplished taking into account the reviewer’s reports.

1.- The manuscript was generally good, there are some grammatical and syntax errors in the text. Thus, the language of paper has to be checked carefully again.

R1: Thank you very much for your positive comments. English has been checked by an expert professional linguist. Christine Francis (UK) who works with us on this aspect has carried out the review.

2.- Abbreviations used in whole manuscript have to be defined firstly and then their abbreviations have to be used.

R2: Abbreviations had been defined in the original version of this manuscript (Line 29, 110, 112).

3.-  The abstract of the MS is not well documented to represent the whole manuscript objectives. Thus, it should be rewritten again in order to represent the paper sufficiently.

R3: Thank you very much. Abstract has been rewritten to represent the complete objectives as the Reviewer suggested.

4.- Regarding to the originality of the manuscript in the last paragraph (Introduction), it should be indicated clearly how this manuscript contributes to the existing knowledge. Please also identify and describe the originality of the study described in this manuscript and how it will likely to contribute to the state-of-the-art. Also, The introduction should be enrich by update references in the field of dairy products.

R4: Thank you very much. In the last paragraph of the Introduction, we have specified the contribution of this manuscript to the existing knowledge and we have described the novelty of the present study in the present scientific context. Some references were included in the introduction. Line 72-77.

Jafarzadeh, S.; Salehabadi, A.; Nafchi, A.M.; Oladzadabbasabadi, N.; Jafari, S.M. Cheese packaging by edible coatings and biodegradable nanocomposites; improvement in shelf life, physicochemical and sensory properties. Trends Food Sci. Technol., 2021, 116, 218-231,

Paidari, S.; Ahari, H.; Pasqualone, A.; Anvar, A.; Allah Yari Beyk, S.; Moradi, S. Bio-nanocomposites and their potential applications in physiochemical properties of cheese: an updated review. J. Food Meas. Charact., 2023, in press.

5.- Conclusion part is short, should be revised. It should contain main results and suggestions for further or future studies.

R5: Thank you very much. Conclusions have been completed with the principal results and possible new studies to extend our knowledge of San Simón da Costa cheese.

6.- What about the microbiological analysis for the cheese during storage. I advise to  determine the microbiological load for the samples during storage that related also to the ripening of cheese.

R6: Thank you very much. The authors agree that a microbiological study would complete the study performed. It is obvious that in cheese there are many interactions between microorganisms in relation to flavor formation. We know through different studies the evolution of the different microbial groups throughout the maturation process (García et al., 2002; Garabal et al., 2008; Rodríguez-Alonso et al., 2008; García-Fontán et al., 2001). On the other hand, as mentioned in our article, there is already a previous microbiological study when evaluating the effect of vacuum packaging and different atmospheres during storage (Garabal et al., 2010). Due to the length of the paper we have not included this aspect.

Garabal, J.I.; Rodríguez-Alonso, P.; Centeno, J.A. Chemical and biochemical study of industrially produced San Simón da Costa smoked semi-hard cow’s milk cheese: effects of storage under vacuum and different modified atmospheres. J. Dairy Sci., 2010, 93, 1868–1881.

M.C. Garcı́a, M.J Rodrı́guez, A. Bernardo, M.E Tornadijo, J. Carballo. Study of enterococci and micrococci isolated throughout manufacture and ripening of San Simón cheese. Food Microbiol., 2002, 19, 23-33. https://doi.org/10.1006/fmic.2001.0457

J.I. Garabal, P. Rodríguez-Alonso, J.A. Centeno. Preliminary characterization of lactic acid bacteria isolated from raw cows’ milk cheeses currently produced in Galicia (NW Spain). LWT Food Sci. Technol., 2008, 41, 1452-1458. https://doi.org/10.1016/j.lwt.2007.09.004.

  1. Rodríguez-Alonso, J.I. Garabal, J.A. Centeno. Preliminary characterization of staphylococcal, micrococcal and yeast isolates obtained from raw cows’ milk cheeses currently produced in Galicia (NW Spain). Ital. J. Food Sci., 2008, 2, 161-169.

M.C. García-Fontán, I. Franco, B. Prieto, M.E. Tornadijo, J. Carballo. Microbiological changes in “San Simón” cheese throughout ripening and its relationship with physico-chemical parameters. Food Microbiol., 2001, 18, 25-33. https://doi.org/10.1006/fmic.2000.0351

7.- Statistical analysis – the number of trials and repetitions is not mentioned; confidential interval not mentioned; test used for differences between samples – not mentioned.

R7: The number of trials and repetitions had been mentioned in the initial version please, see in the original manuscript: Line 119 (number of trials for chemical and physico-chemical analysis). Lines 143-144 (number of trials for puncture tests), Lines 152-153 (number of trials for viscoelastic tests). Moreover, confidential intervals and test used for differences between samples had been mentioned in 2.9 section (Lines 193-202) in the original manuscript. These aspects have been further clarified in the current version.

  1. the sensory evaluation should be done by expert persons and how many person was evaluated the sensory of cheese?

R8: Thank you very much. In the original version of the manuscript we have pointed up that sensory evaluation has been done by ten persons (20-50 years old) which formed the trained panellists. All members responsible for judging the sensory quality of Galician industrial cheeses with PDO status.

Reviewer 3 Report

This paper aims to present the effect of vacuum packaging on traditional Spanish cheese, the San Simón da Costa. The paper presented a lot of results on the variation of biochemical, physicochemical and sensory properties of the cheeses during storage.  However, the manuscript presented some drawbacks and needs to be improved deeply before being considered for publication in Foods journal. The paper needs a strengthened experimental design and modification of the paper format.

- Principal remarks:

-          Regarding the experimental design, the paper presents only one packaging condition and then proposes a multitude of measures on this condition (one plastic material for packaging coupled with vacuum ‘’atmosphere’’). We invite the authors to propose a broader experimental design by testing different packaging materials generally used for cheeses or at least for traditional cheeses.

-          A very large number of illustrations (especially tables – 10 tables) are presented in the paper. Some table can be easily merged to make the reading easier.

-          Captions of tables are often incomplete and push the reader to look for the meaning of the abbreviations in the text.

- Some detailed comments are listed below:

+ In the abstract:

-          The first sentence needs to me reformulated or deleted. This is right only for traditional cheeses and not for the others and more particularly for PDO cheeses.

-          Please reformulate the two first sentences of the abstract.

-          Correct the last sentence of the abstract.

+ Introduction:

-          Lines 81-87. Please add references to this part.

-          Lines 90-93. If this cheese has already a long life, what is the interest to use vacuum packaging? Please be clearer in this part concerning the interest of using vacuum packaging.

+ Materials and methods

-          Line 110. Clear “rheology”

-          Lines 112 and Line 114: n=12 – what is the meaning?

-          Line 155. This sentence is not necessary

+ Results and discussion

-          Table 1. The cheese names can be written and not the abbreviations. This will simplify the reading and understanding of the results.

-          Table 1. Put in the first column the cheese names

-          Some factors presented in the table 1 are not described and discussed (TA, Ash, aw...). Please discuss those results.

-          Line 254-256. Give some hypothesis to explain those differences.

-          Lines 259-260, please correct/reformulate the sentence. This is not observed in the results presented.

-          Line 267. Reformulate.

-          Lines 281-285. This can be discarded from this part and added in the next part of the paper when the rheology and texture results are discussed.

-          Lines 304-309. This needs to be more discussed. A PCA can help to deeply discuss those results. We encourage the authors to use this analytical tool.

-          Table 4. Why the results of the cheeses properties for the V samples time 0 are presented in tables 3 and 4 and not in the other tables?

-          Line 354-355. This is in contradiction with the discussion presented in lines 306-309, no?

Author Response

Authors want to thank both referees for review. In our opinion, their comments, suggestions, and criticisms really improved the manuscript. According to the editor’s recommendation, a revision to the original version was performed. This was mainly accomplished taking into account the reviewer’s reports.

1.- This paper aims to present the effect of vacuum packaging on traditional Spanish cheese, the San Simón da Costa. The paper presented a lot of results on the variation of biochemical, physicochemical and sensory properties of the cheeses during storage.  However, the manuscript presented some drawbacks and needs to be improved deeply before being considered for publication in Foods journal. The paper needs a strengthened experimental design and modification of the paper format.

R: Thank you very much for your comment. We have trained to strengthen the experimental design. Nevertheless, further research should be carried out to determine the impact of new biomaterials on cheese coating and packaging. Format of paper was the same than that proposed by the Journal’s template. 

2.- Regarding the experimental design, the paper presents only one packaging condition and then proposes a multitude of measures on this condition (one plastic material for packaging coupled with vacuum ‘’atmosphere’’). We invite the authors to propose a broader experimental design by testing different packaging materials generally used for cheeses or at least for traditional cheeses.

R: Thank you very much for your comment. We agree with the Reviewer on the scientific interest of testing with diverse packaging materials for cheeses. The present study was based on the common plastic bags used commercially as vacuum packing material. The purpose was to give scientific and practical information to the Regulation Council of San Simón da Costa PDO comparing the physiochemical, mechanical, viscoelastic and sensory characteristics of natural and the commonly vacuum-packed cheeses. Other studies may be done to compare different packaging materials to distinguish their influence on the textural quality of this cheese.

San Simón cheese is smoked after 45 days of ripening. The aroma in cheeses packaged under vacuum or in different atmospheres is negatively evaluated. The use of biopolymers (as mentioned in the introduction and conclusions) could be an excellent approach for future research.

3.- A very large number of illustrations (especially tables – 10 tables) are presented in the paper. Some tables can be easily merged to make the reading easier.

R: Thank you very much for your comment. The number of tables was reduced to simplify the manuscript. Tables 7, 8 and 9 were removed and the results discussed in the related section.

4.- Captions of tables are often incomplete and push the reader to look for the meaning of the abbreviations in the text.

R: Thank you very much for your comment. Caption of Tables has been completed including the meaning of abbreviations.

+ In the abstract:

5.- The first sentence needs to me reformulated or deleted. This is right only for traditional cheeses and not for the others and more particularly for PDO cheeses.

R: Thank you very much for your comment. The first sentence was deleted to simplify the abstract.

6.- Please reformulate the two first sentences of the abstract.

R: Thank you very much for your comment. The two first sentences were reformulated as the Reviewer suggested.

7.- Correct the last sentence of the abstract.

R: Thank you very much for your comment. The last sentence of the abstract was corrected.

+ Introduction:

8.- Lines 81-87. Please add references to this part.

R: Thank you very much for your comment. We have introduced the reference you recommend and another article with a new revision.

  • Jafarzadeh, S.; Salehabadi, A.; Nafchi, A.M.; Oladzadabbasabadi, N.; Jafari, S.M. Cheese packaging by edible coatings and biodegradable nanocomposites; improvement in shelf life, physicochemical and sensory properties. Trends Food Sci. Technol., 2021, 116, 218-231,
  • Paidari, S.; Ahari, H.; Pasqualone, A.; Anvar, A.; Allah Yari Beyk, S.; Moradi, S. Bio-nanocomposites and their potential applications in physiochemical properties of cheese: an updated review. Food Meas. Charact., 2023, in press.
  •  

9.- Lines 90-93. If this cheese has already a long life, what is the interest to use vacuum packaging? Please be clearer in this part concerning the interest of using vacuum packaging.

R: Thank you very much for your comment. It has been included a sentence to explain the technological interest of the vacuum packing in this kind of cheese. (Line 90-93).

+ Materials and methods

10.-  Line 110. Clear “rheology”

R: Thank you very much for your comment but we did not find this word neither in the line you mention nor in the adjacent lines.

11.- Lines 112 and Line 114: n=12 – what is the meaning?

R: Thank you very much for your comment. It has been clarified in the manuscript. Four batches were used and four cheeses from each batch. A total of 12 cheeses were vacuum packed and another 12 remained unpacked under the same environmental conditions.

12.- Line 155. This sentence is not necessary

R: Thank you very much for your comment. The sentence has been removed.

+ Results and discussion

13.- Table 1. The cheese names can be written and not the abbreviations. This will simplify the reading and understanding of the results.

R: Thank you very much for your comment. The cheese names were completely written and abbreviations were removed from Table 1 as the Reviewer suggested. The same criterion was extended to the other Tables. 

14.- Table 1. Put in the first column the cheese names

R: Thank you very much for your comment. The cheese names were included in the first column as the Reviewer suggested. 

15.- Some factors presented in the table 1 are not described and discussed (TA, Ash, aw...). Please discuss those results.

R: Thank you very much for your comment. TA, ash and aw were described in Table 1 and discussed in the manuscript (Line 244-253).

16.- Line 254-256. Give some hypothesis to explain those differences.

R: Thank you very much for your comment. We have included a reference to previous work for clarification. The explanation would also be provided in the following section (line 276-278) where it is mentioned that “It therefore seems that chilled storage (4 °C) will slow down the enzymatic action, which is consistent with the low intensity of proteolytic reactions observed at this stage.

17.- Lines 259-260, please correct/reformulate the sentence. This is not observed in the results presented.

R: Thank you very much for your comment. A reference to a previous work has been introduced to clarify the sentence. Line 274.

18.- Line 267. Reformulate.

R: Thank you very much for your comment. That line has been reformulated as the Reviewer suggested. Line 279-281.

19.- Lines 281-285. This can be discarded from this part and added in the next part of the paper when the rheology and texture results are discussed.

R: Thank you very much for your comment. Those Lines have been moved to the rheology part as the Reviewer suggested. Line 326-331.

20.- Lines 304-309. This needs to be more discussed. A PCA can help to deeply discuss those results. We encourage the authors to use this analytical tool.

R: Thank you very much for your comment. We agree with the Reviewer on a PCA is an excellent analytical tool. In this case, in order to simplify the discussion, we consider that the correlations used clarify the results obtained well.

21.- Table 4. Why the results of the cheeses properties for the V samples time 0 are presented in tables 3 and 4 and not in the other tables?

R: Thank you very much for your comment. The Reviewer is right. Tables 3 and 4 have been changed to follow the same format than the others. Obviously at t=0 there are not differences between natural and vacuum-packed cheeses.

22.- Line 354-355. This is in contradiction with the discussion presented in lines 306-309, no?

R: Thank you very much for your comment. No, there is no any contradiction. So, in lines 354-355 (puncture tests) we have focussed the analysis on the influence of storage time on BD values for 0-2 months, when the differences were significant (p<0.05) for V samples. We did not consider BD data between 2 and 6 months because BD were similar. However, in lines 306-309 (mechanical spectra in the LVER) the methodology of analysis was different, because we compared simultaneously the effect of storage time and the vacuum packing on cheeses in the complete time storage (0-6 months) when the viscoelastic parameters were noticeable different. In both Puncture and oscillatory data the trends were analogous since N and V cheeses were strengthened during storage time being the effect more intense in N than V samples.

Round 2

Reviewer 1 Report

All my comments were addressed fairly.

Author Response

We thank the reviewers for their detailed reading and positive response to our manuscript.

The authors would like to thank both reviewers. In accordance with the editor's recommendation, a revision of the original version was carried out. Attached is a letter from Christine Francis who usually works with us to carry out the English revision.

Reviewer 3 Report

Dear Editor,

All the comments have been adressed. Nonetheless, I still have a small comment concerning the abstract : It is not clear what high "high S" parameter?

Best regards.

Author Response

We thank the reviewers for their detailed reading and positive response to our manuscript.

The authors would like to thank both reviewers. In accordance with the editor's recommendation, a revision of the original version was carried out. Attached is a letter from Christine Francis who usually works with us to carry out the English revision.

Thank you for your comment on the parameter S. We have clarified this concept in the abstract.

We have updated our manuscript in response to comments from all reviewers and Editor.
